# An Eye for an Ear:
# Zero-shot Audio Description Leveraging an Image Captioner using Audiovisual Distribution Alignment

**Hugo Malard**[1]    **Michel Olvera**[1]    **Stéphane Lathuiliere**[1]    **Slim Essid**[1]

[1]LTCI, Télécom Paris, Institut Polytechnique de Paris

{hugo.malard, michel.olvera}@telecom-paris.fr

## Abstract

Multimodal large language models have fueled progress in image captioning. These models, fine-tuned on vast image datasets, exhibit a deep understanding of semantic concepts. In this work, we show that this ability can be re-purposed for audio captioning, where the joint image-language decoder can be leveraged to describe auditory content associated with image sequences within videos featuring audiovisual content. This can be achieved via multimodal alignment. Yet, this multimodal alignment task is non-trivial due to the inherent disparity between audible and visible elements in real-world videos. Moreover, multimodal representation learning often relies on contrastive learning, facing the challenge of the so-called modality gap which hinders smooth integration between modalities. In this work, we introduce a novel methodology for bridging the audiovisual modality gap by matching the distributions of tokens produced by an audio backbone and those of an image captioner. Our approach aligns the audio token distribution with that of the image tokens, enabling the model to perform zero-shot audio captioning in an unsupervised fashion while keeping the initial image captioning component unaltered. This alignment allows for the use of either audio or audiovisual input by combining or substituting the image encoder with the aligned audio encoder. Our method achieves significantly improved performances in zero-shot audio captioning, compared to existing approaches.[1]

## 1  Introduction

Recent progress in image captioning has been driven by methods integrating Large Language Models (LLMs) with vision encoders (1; 2; 3). The impressive capabilities of Vision Language Models (VLMs) stem from *supervised* training on large image-text collections and extensive parameterization. Even the smallest VLMs exceed more than a billion parameters (4; 5). These models excel in generating human-like text descriptions for images and understanding complex semantic relationships. Such capabilities can be potentially extended to other tasks, beyond image analysis.

One such task is audio captioning, in which large-scale audio-text collections, with precise descriptions of the audio content, are lacking. Recent works have proposed diverse solutions to address this limitation. Solutions include augmenting class labels with phrases such as "*This is a sound of*" (6; 7) or prompting LLMs to generate natural language descriptions directly from class labels (8). Further, datasets from related domains like Speech (9) or Music (10; 11), have been considered to enlarge training data. While such methods show impressive results, scaling them further is challenging due to the rarity and low quality of such labels-captions. In particular, these approaches cannot cope with the annotation noise encountered in existing audio datasets, especially in the largely adopted resource: audio tracks of AudioSet (12), which is only partially and weakly labeled.

---

[1]https://github.com/hugomalard/AnEyeForAnEar.git

38th Conference on Neural Information Processing Systems (NeurIPS 2024).

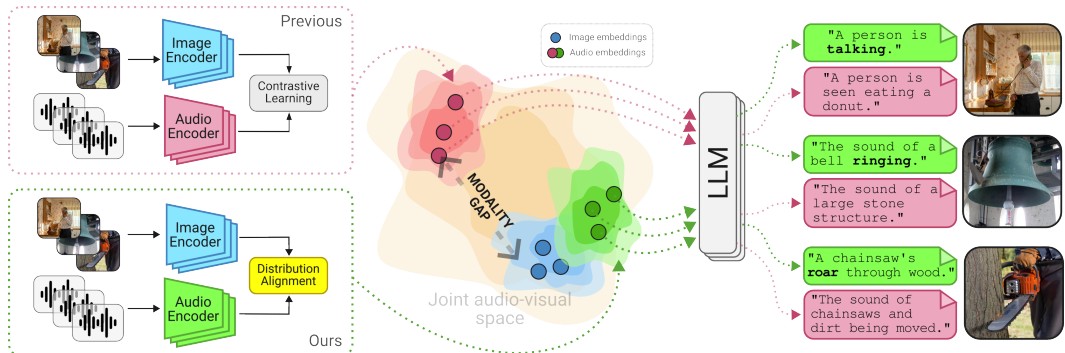

Figure 1: Conventional audiovisual alignment through contrastive learning leads to a gap between modalities. Our proposed distribution alignment method matches closely both distributions leading to better joint representations for audio captioning.

We hypothesize that VLM-based image captioning models, inherently possess the knowledge to perform audio captioning. Thus, joint image-language representations of VLMs can be used to describe auditory content associated with image sequences from videos featuring audiovisual content, eliminating the need for handcrafted audio-caption pairs. To our knowledge, this is the first attempt to leverage VLMs for this purpose. Extending VLMs to audio captioning is non-trivial due to the inherent disparity between audible and visible elements in real-world videos. For instance, sound-producing objects in images may be occluded or out of view, while visible objects might not generate sound. This mismatch hinders connecting visual and auditory modalities seamlessly.

Multimodal representation learning (13), crucial for tasks like image captioning, is often accomplished through contrastive learning (14; 6). This paradigm aligns features across modalities by maximizing agreement between similar samples and minimizing agreement between dissimilar ones. Despite its effectiveness, contrastive learning faces the so-called "modality gap", where embedded data are represented in distinct, non-overlapping regions of the embedding space (15; 16) (the distance between red and green manifolds in Figure 1), limiting flexibility when conditioning modalities.

We propose a novel methodology to bridge the audiovisual modality gap by matching the distributions of tokens produced by an audio backbone and those from the encoder of an image captioner (represented as the green and blue manifolds in Figure 1), enabling the image captioner to perform **zero-shot** audio captioning in an **unsupervised** way. We define *zero-shot audio captioning* as the task of generating audio captions without training on manually assembled audio-caption pairs. We propose and compare two token distribution alignment methods: one based on Maximum Mean Discrepancy (MMD) (17) and a more flexible alternative using Optimal Transport (OT) (18), possibly enhanced with a cross-attention mechanism. This mechanism refines distribution matching by attending to each token within a modality and determining its match with a token from the other modality based on semantic similarity. We find this approach to result in more accurate and coherent joint audiovisual representations.

Acknowledging the challenges of zero-shot audio captioning, we enhance caption quality using "prefix tuning", a technique commonly employed across domains to adapt LLMs to diverse tasks in a few-shot manner (19; 20; 21). This tuning process conditions the model with few image-audio caption pairs, prompting it to generate audio-centric captions and thereby boosting performance across standard audio captioning metrics (22; 23; 24; 25; 26; 27).

Notably, our methodology allows for the use of either audio or audiovisual input by combining or substituting the image encoder with the aligned audio encoder. As a result, our approach extends the ability of the VLM to audio captioning without compromising its image performance. Moreover, the method achieves significantly improved performances in zero-shot audio captioning, compared to existing approaches.

In summary, our contributions are as follows:

- We introduce a novel methodology for unsupervised zero-shot audio captioning through Distribution ALIgnment (DALI), leveraging advanced image captioning models, and suc-

cessfully instantiate it with a particular image captioner (namely Llava 1.5), as we obtain state-of-the-art zero-shot audio captioning results without any supervision from annotated audio data (see Section 5).

- To bridge the "modality gap", we propose and study two multimodal token distribution alignment methods exploiting MMD or Optimal Transport (OT). The latter—never used before in this context—is enhanced with a cross-attention mechanism for greater robustness.

- We introduce the use of prefix tuning to guide the image captioner towards audio captioning, adapting the model with a few shots of image and audio-caption pairs (only using the textual descriptions of sound and not the audio signals).

- Finally, our method supports both audio and audiovisual inputs, extending the image captioner's capabilities without compromising its performance in image analysis tasks.

## 2   Related works

**Vision Language models**   Large Vision Language Models (VLMs) typically use a pre-trained LLM that processes both visual and textual information. The image encoder usually consists of a ViT (28) trained via multimodal contrastive learning using millions of image-text pairs such as the ones used in CLIP (14). Integrating visual information into the LLM is usually done by concatenating image tokens extracted by a vision encoder and then further transformed by an MLP, with textual tokens from a prompt. This straightforward architecture can be easily adapted to other modalities by simply replacing the input tokens to accommodate the new input modality.

**Audio Language models**   Recently, a similar paradigm emerged in the audio application domain. Various works rely on both LLMs and audio encoders trained jointly on millions of audio-text pairs, similarly to CLAP (6). Listen Think and Understand (LTU) (29), Pengi (11), QwenAudio (9) and AudioFlamingo(10) rely on a language model fed with tokens from both audio and text modalities. These large models require substantial volumes of training data. For instance, AudioFlamingo has been trained on nearly all publicly available labeled audio data. One limitation is that the quality of the audio-text data is generally not comparable to that of image-text data. This situation is made worse by the fact that audio data labelling is inherently difficult due to the potential overlap of multiple sound sources at different acoustic levels, which sometimes makes it hard to perceive some of the audio classes in presence. Despite this limitation, AudioFlamingo and similar models exhibit remarkable performance. To further enhance the capabilities of audio-language models it is of paramount importance to explore alternative paradigms beyond strong reliance on (often unreliable) annotations and supervised learning. This is the goal pursued in this work.

**Audio captioning**   Training large audio captioner models, as mentioned previously, requires large volumes of audio and textual descriptions and substantial computational resources. To address this limitation, some approaches leverage existing robust backbones to enable zero-shot audio captioning. ZerAuCaps (30) uses CLAP (an 80.8M-parameter model trained on 3.4M audio-text pairs) to infer potential classes from a predefined word bank (45 words) before using them to prompt an LLM (OPT (31) 1.3B), generating multiple plausible captions. The final caption is the closest to the audio in the CLAP space. However, this method is constrained by the finite set of classes in the word bank, limiting its application to predefined scenarios. The most closely related work to ours is that of Shaharabany et al. (32) who also performed unsupervised zero-shot audio captioning. The authors adopted the audio encoder of ImageBind (33) (86M parameters trained on AudioSet), a multimodal model trained to align various modalities in the image space via a contrastive loss. They fine-tuned a GPT-2 (34) model (117M parameters) fed with ImageBind audio tokens using two distinct loss functions for the captioning task: an audibility score ensures the generation of audio-centric captions without visual artifacts, while an ImageBind score measures the similarity between the generated caption and the audio. Despite its innovative approach, its performance falls short compared to ZerAuCaps primarily due to ImageBind's less effective handling of audio.

**Audiovisual Alignment**   Multimodal alignment aims to encode information from different modalities such as text, image, or audio into a shared semantic space, enabling cross-modal relationships and multimodal retrieval (*e.g.*, retrieve the image closest to a caption) (13). Despite the abundance of web video datasets, aligning the audio and image/video modalities remains a challenge. The conventional contrastive loss (35) often fails to effectively co-train the encoders. Unlike the relatively clean associations in image-text pairs, audio-image pairs present a greater degree of variability, with

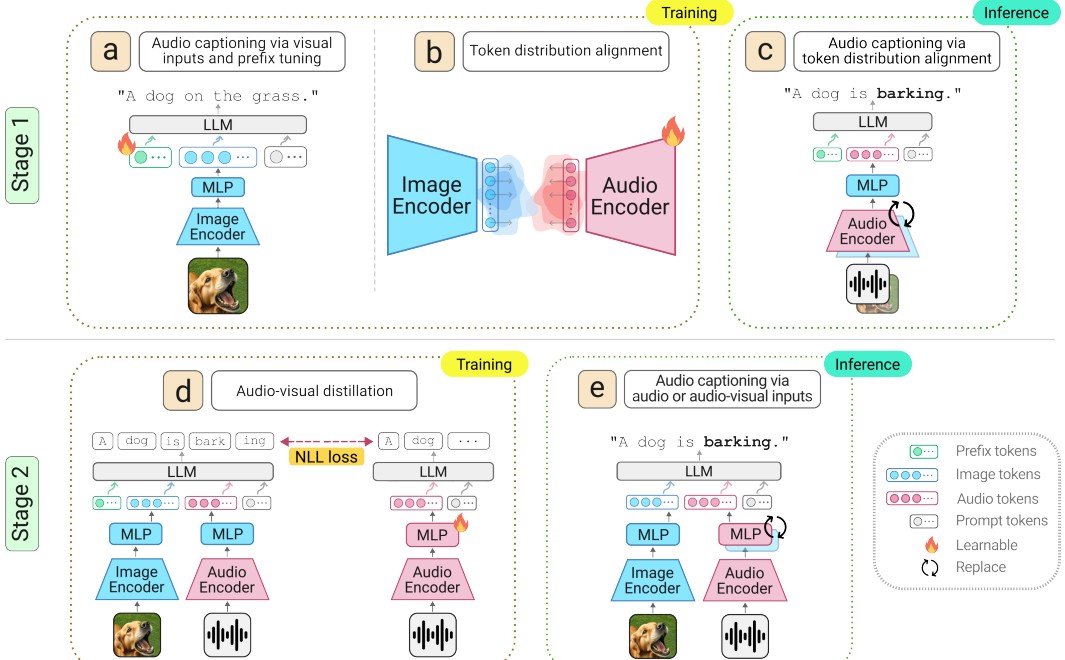

Figure 2: Overview of the proposed approach. In the first stage, a prefix tuning is performed using a few (image, audio-caption) pairs (1-a). Additionally, the audio backbone is aligned with the image backbone (1-b) through distribution alignment. Audio captioning can then be performed by switching the image backbone with the audio backbone and adding the prefix tokens (1-c). In a second stage, visually-informed audio captions are generated using both audio, image, and prefix tokens. The MLP mapping the audio encoder to the language model is then fine-tuned with these pseudo captions (2-d). The final inference for audio captioning, using audio or audio visual inputs, is performed by forwarding the aligned audio backbone's output through the trained MLP to obtain the LLM input (2-e).

the content described in the audio often differing from that depicted in the corresponding image or video. Consequently, stable pre-training needs augmenting the contrastive loss with a reconstruction loss (36; 37; 38), reducing the similarity of the representations between the two modalities.

# 3   Method

Current Large Vision Language Models (VLMs) exhibit a sophisticated understanding of concepts related to real-world objects in the visual domain. This capability arises from optimizing the language model, already containing a lot of intrinsic knowledge, so as to extract additional information from image features and learn how to describe the (visual) world. We hypothesize that the knowledge gained through this process is sufficiently general to be transferable across different modalities. To test this hypothesis, we developed a methodology that employs an VLM's image captioner to generate audio captions from images, audios and audiovisual inputs, **while preserving the original performance of the image captioner.** Our methodology stands out for its applicability to any modality requiring integration with image data. Here, we focus on audiovisual alignment for audio captioning, benefiting from the abundance of audiovisual content in videos from web data.

## 3.1   Proposed Framework Overview

An image captioning system outputs descriptions of objects in an image, such as "A dog in the grass," providing visual cues. To adapt this system for audio captioning, we need it to focus on audible information instead of visuals. For example, given an image of a dog, the audio caption should instead be "A dog is barking.". Our methodology enables audio captioning from an image captioner by keeping both the image encoder and LLM frozen, ensuring that the image captioner original

performance is preserved. In our work, we choose Llava (1) (Apache License 2.0) as an instance of an image captioner, for its proven effectiveness. We use the CAV-MAE (36) (BSD 2-Clause License) audio model as our audio encoder, since it is already roughly aligned with an image backbone, which is expected to help reach faster convergence.

As seen in Figure 2, we follow a 2-stage process. First, we use prefix tuning to start re-tasking the system for audio captioning, learning tokens from a few image-caption pairs, ensuring that the textual descriptions target the audio content (stage 1, step (a)). We also align the token distribution of an audio encoder with that of the image encoder (stage 1, step (b)), then replace the image encoder with the aligned audio encoder for direct audio captioning from audio files (step (c)). We can also combine both encoders for visually informed audio captioning, using both audio and visual inputs. This setup generates audio captions to fine-tune the MLP that maps audio tokens to the language model embedding space, a process we call audiovisual distillation (stage 2, step (d)). The fine-tuned MLP then replaces the original MLP, generating robust audio captions from audio or audiovisual inputs (step (e)). In the following, we zoom on specific aspects of the proposed methodology.

**Prefix-tuning with image-caption pairs**    Image captioners, while effective for image analysis tasks, struggle to produce accurate audio-centric descriptions when prompted to describe sounds from images. For example, when asked, "What sounds can the objects in this image generate?", the descriptions provided heavily reflect the visual characteristics of the objects. Instead of focusing on sound-related details, their outputs are dominated by visual descriptions, which are considered artifacts in the context of audio captioning. To address this problem, we adopted a prompt tuning strategy (39), aiming to guide the model to generate captions that exclusively describe audible events as depicted in Figure 2-a. Prompt tuning involves learning a portion of the prompt tokens: the prefix, that guides the model to perform a specified task (39). This method allows the rest of the model to remain frozen, and by simply removing the learned tokens, the model is able to perform its original task with the same performance. This approach requires only a small number of parameters to be trained and can be effectively trained using a few examples, a process akin to **few-shot adaptation**. We employed this technique to re-task the model to perform audio captioning. We trained the additional tokens using image-text pairs, with the text describing the sounds associated with the images *i.e.*, audio captions. Specifically, we used images from AudioSet videos and corresponding captions from AudioCaps (40) (MIT). We used negative log-likelihood as the loss function for the prefix tuning:

$$\mathcal{L}_{\text{prefix}} = -\sum_{n=1}^{N} \sum_{j=1}^{l} \log p_\theta(C_j^n | I_1^n, ..., I_{n_i}^n, C_1^n, ..., C_{j-1}^n) \; ; \qquad (1)$$

with $n$ and $j$ being respectively the image and caption token indices, $N$ the number of elements in the batch, $l$ the number of predicted tokens, the index $I$ referring to the vision tokens (from 1 to $n_i$), $C_j$ the $j$-th caption token and $\theta$ the parameters of the model. Using the prefix tokens, the image captioner can now output more plausible sound descriptions.

**Audio captioning via token distribution alignment**    Aligning the distribution of the audio encoder with that of the image one allows for substituting one encoder with another and therefore conditioning the generation either on the audio or image modality. We align the distributions (Figure 2: step (b)) using either Maximum Mean Discrepancy or Optimal Transport. More details are provided in Section 3.2. Using the aligned audio backbone, as represented in Figure 2: step (c), we feed the language model with the audio tokens along with the prefix, which enables the model to perform audio captioning. Additionally, we incorporate both the image and audio tokens, to give the ability to the model to perform audio captioning fusing visual and audible cues. This setup allows us to infer and generate, visually informed, audio captions for AudioSet balanced (12), a standard subset of AudioSet, containing $\sim$ 20k videos, curated such that they involve the same number of samples for each class.

**Audiovisual distillation**    The captions generated from audiovisual inputs serve as pseudo-labels to supervise the fine-tuning process of an audio captioner with an audio-only setup. This process refines only the MLP that maps the audio tokens to the LLM embedding space. Training employs a negative log-likelihood loss, as per equation (1), albeit without conditioning on image tokens. Instead of relying on a limited number of real captions, pseudo-labels are employed. This procedure is

illustrated in Figure 2: step (d), and optimized through the following loss:

$$\mathcal{L}_{\text{distil}} = -\sum_{n=1}^{N} \sum_{j=1}^{l} \log p_\theta(C_j^n | A_1^n, ..., A_{n_a}^n, C_1^n, ..., C_{j-1}^n) \; ; \tag{2}$$

with $A$ referring to the audio tokens (from 1 to $n_a$).

**Audio captioning via audio or audiovisual inputs**    With all components trained, our model can perform audio captioning using the aligned audio encoder and fine-tuned audio MLP, or visualy-informed audio captioning by feeding to the LLM the image tokens in addition to the audio tokens (shown in Figure 2: step (e)). Notably, the original image captioning task can still be performed at its original performance level by using only the image tokens, without the audio.

## 3.2 Token distribution alignment

Many studies on multimodal alignment typically use a contrastive loss on the averaged output tokens (14; 41). This method merely aligns token mean-statistics, potentially leading to information loss, when the full token distribution is needed. Llava (1) for instance, makes use of the full token distribution as input, naturally creating the need for a full token distribution alignment, to allow for swapping the pretrained encoder with a new one targeting a new modality. Moreover, contrastive learning faces the so-called "modality gap" problem where each modality is encoded in distinct sub-regions of the embedding space (see Figure 4). For all these reasons, replacing image tokens with audio tokens obtained by alignment through a contrastive learning approach may yield undesirable responses from the language model. In light of these issues, we propose to learn the full image token distributions, rather than relying solely on contrastive learning, so as to facilitate knowledge transfer. We refer to our distribution alignment framework as DALI which stands for Distribution ALIgnment. We study two variants of distribution matching.

**Modality alignment through MMD**    Maximum Mean Discrepancy (MMD) serves as a robust measure of dissimilarity—or more specifically, discrepancy—between two probability distributions. MMD has been shown to be effective in distribution alignment, notably, but not exclusively, for domain adaptation (42; 43). It computes the distance between the expected values of two distributions, denoted by $P$ and $Q$, within a feature space characterized by a mapping $\Phi : X \rightarrow \mathcal{H}$ where $\mathcal{H}$ is a reproducing kernel Hilbert space (RKHS):

$$\text{MMD}(P, Q) = ||\mathbb{E}_{X \sim P}[\Phi(X)] - \mathbb{E}_{Y \sim Q}[\Phi(Y)]||_{\mathcal{H}} \; . \tag{3}$$

In practice the MMD is computed using the kernel trick (44). In our work, we consider the Gaussian Radial Basis Function (RBF) kernel and use MMD as a loss function to directly align the audio and image token distributions. We term this distribution alignment through MMD: DALI$_{\text{MMD}}$.

**Modality alignment through Optimal Transport**    Optimal Transport (OT) stands out as a powerful method for aligning probability distributions (45). Despite its successful application across various domains such as unsupervised domain adaptation (46), image generation (47), and style transfer (48), its application to multimodal representation alignment remains relatively unexplored. In our context, where the alignment of audiovisual modalities is paramount, OT emerges as a natural solution for its intrinsic capacity to integrate the geometry of the underlying space, enabling optimal distribution matching. Discrete optimal transport considers two distributions $x \in \mathbb{R}^N$ and $y \in \mathbb{R}^M$, where $N$ and $M$ represent the number of samples in each distribution. These distributions are represented by their empirical measures: $\alpha = \sum_{i=1}^{N} \alpha_i \delta_{x_i}$ and $\beta = \sum_{j=1}^{M} \beta_j \delta_{y_j}$, and OT seeks a coupling $\gamma \in \Pi(\alpha, \beta)$ between them that minimizes a transportation cost. The problem can be formalized as:

$$\text{OT}(\alpha, \beta) = \min_{\gamma \in \Pi(\alpha,\beta)} \langle \gamma, D \rangle_{\text{F}} \; ; \tag{4}$$

where $D \in \mathbb{R}^{N \times M}$ is the cost matrix, $\langle \cdot, \cdot \rangle_{\text{F}}$ denotes the Frobenius dot product, and $\gamma \in \mathbb{R}^{N \times M}$ represents the transportation coupling matrix. The minimum of the optimization problem can be interpreted as a distance (45). When considering the square of the $l_2$ norm, the distance is known as the Earth Mover Distance (EMD). In this work, we align the audio and image token distributions using EMD and refer to this approach as Distribution ALIgnment through Optimal Transport (DALI$_{\text{OT}}$).

**Improving cross-modal correspondence with attentive distribution alignment**  Optimal transport is typically applied assuming uniform distributions of the weights $\{\alpha_i\}_{i=1}^N$ and $\{\beta_j\}_{j=1}^M$. However, in audiovisual content, objects producing sound are often occluded, with the visual information serving to complement the auditory content. The classic formulation of OT enforces strict mass preservation, needing that all mass from the source distribution is transported to the target distribution. This constraint becomes problematic in audiovisual alignment, where not all tokens from one modality may find a match in the other. While alternative methods exist to circumvent these constraints, such as Unbalanced Optimal Transport (UOT) (49), which replaces this rigid requirement with a soft penalization term, the tuning of such hyper-parameter poses significant challenges (50). Alternatively, we propose learning the weights $\alpha_i$ and $\beta_j$ via a cross-attention mechanism which offers as well a robust approach. This mechanism, attending to both modalities, enables learning which objects and/or sound-related tokens are present in both modalities. Tokens encoding modality-specific information, e.g., an occluded source, or an object apparently not generating a sound, receive lower weights. Thus, this approach promotes the alignment of modality-shared content. We provide the details of this cross-attention mechanism in Appendix F. It is important to note that to avoid the trivial solution that would put all the weights to 0 except the one of the closest tokens, we regularize the entropy of the weights.

The final loss consists of a weighted sum of the optimal transport distance and the entropy of distribution weights:

$$\mathcal{L}(x, y, \alpha^{\mathrm{Att}}, \beta^{\mathrm{Att}}) = \mathrm{OT}(x, y, \alpha^{\mathrm{Att}}, \beta^{\mathrm{Att}}) - \lambda(\mathrm{H}(\alpha^{\mathrm{Att}}) + \mathrm{H}(\beta^{\mathrm{Att}})), \tag{5}$$

where $\mathrm{H}(\cdot)$ denotes the Shannon entropy and $\mathrm{OT}(x, y, \alpha^{\mathrm{Att}}, \beta^{\mathrm{Att}})$ represents the optimal transport between $\alpha^{\mathrm{Att}}$ and $\beta^{\mathrm{Att}}$, the weights at the output of the attention mechanism, with the cost matrix computed from $x$ and $y$. An intuitive depiction of the method is illustrated in Figure 3. Note that the cross-attention layers are only used for training and discarded afterward. We refer to this alignment method as $\mathrm{DALI}_{\mathrm{OT}}^{\mathrm{Att}}$.

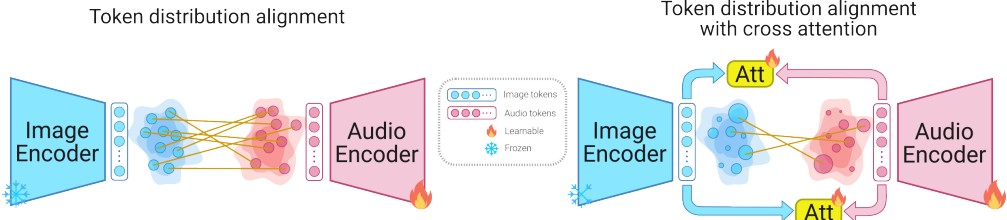

Figure 3: Multimodal distribution alignment through optimal transport. The audio and image tokens are used to compute the cost matrix, while two separate cross-attention layers estimate the weights $\alpha^{\mathrm{Att}}$ and $\beta^{\mathrm{Att}}$.

## 4  Experiments

Our experiments focus on AudioSet, a widely explored large-scale collection of videos from YouTube. We evaluate our proposed approach on two popular and publicly available human-annotated audio captioning datasets: AudioCaps and Clotho (Tampere University licence), using conventional audio captioning metrics. We also provide details of our training and fine-tuning procedures.

**Datasets**  AudioSet (12) is a dataset (license CC BY 4.0) composed of 2 million YouTube videos with annotations indicating the presence of environmental sound events at the clip level (without precise time location), spanning 527 classes. The dataset includes reliability scores for annotations, some below 50%. Despite a diverse range of classes, a significant portion predominantly features music or speech content. Many videos show notable disparities between their visual and auditory content, thus attempting to align audio and image representations without any pre-filtering of the content is challenging. Similarly to Nagrani et al. (51), who filtered the dataset by balancing the classes, we train our models on a subset of 500k videos from AudioSet. We chose them by computing the similarity between the noisy audio labels and an image caption generated by BLIP-2(52) (chosen for the conciseness of the captions), to remove the biggest discrepancies between images and audios. More details on this filtering process are given in Appendix C. For each video, we extracted 10 frames (the first of each second), and we randomly chose, during training, one of these 10 frames for each video.

|  | Alignment | METEOR↑ | ROUGE-L↑ | SPIDEr↑ |
|---|---|---|---|---|
|  | Image only[*] | 0.1324 | 0.3008 | 0.1499 |
| Ours (Audio-Only) | $\text{DALI}_{\text{OT}}^{\text{Att}}$ | 0.1277 | 0.3106 | 0.1592 |
|  | Contrastive | 0.1215 | 0.2914 | 0.1524 |
|  | $\text{DALI}_{\text{OT}}$ | 0.1332 | 0.2923 | 0.1422 |
|  | $\text{DALI}_{\text{MMD}}$ | **0.1346** | 0.3025 | 0.1360 |
| Ours (Audio+Image) | $\text{DALI}_{\text{OT}}^{\text{Att}}$+Image | 0.1257 | 0.3061 | **0.1946** |
| Shaharabany et al. | ImageBind | n.a | 0.082 | n.a |
| Salewski et al. | CLAP[**] | 0.123 | **0.331** | 0.183 |
| Ablation of audiovisual distillation (Stage 2-d) | $\text{DALI}_{\text{OT}}^{\text{Att}}$ | 0.11 | 0.2901 | 0.1443 |
|  | $\text{DALI}_{\text{MMD}}$ | 0.1330 | 0.3018 | 0.1385 |
|  | $\text{DALI}_{\text{OT}}$ | 0.1062 | 0.2709 | 0.1183 |
|  | Contrastive | 0.0728 | 0.1838 | 0.0871 |

Table 1: Audio captioning performance on AudioCaps test set. Our results are obtained using 16 (image,audio-captions) pairs for the prefix tuning phase. (*): No alignment. (**): Trained in a supervised fashion using audio-caption pairs. Results are ordered by SPIDEr score.

**Evaluation metrics**   We evaluated our system on all known audio captioning datasets which are publicly available: AudioCaps (40) which contains AudioSet audio samples captioned by humans, and Clotho (53), which is composed of audio samples from FreeSound, also annotated by humans. Performance was measured using 3 standard captioning metrics: METEOR (24), ROUGE-L (23), and SPIDEr (27), which combines both the CIDEr (25) and SPICE (26) scores. Additional evaluation metrics are available in Appendix G. We evaluate our model in the context of zero-shot audio captioning, comparing it against other models in the field. All the training details and hyper parameters are given in Appendix E.

## 5   Results and discussion

**Audio Captioning performance**   Table 5 presents the audio captioning performance of our methods on the AudioCaps test set, using 16 image captions to train the prefix tokens. We show the relatively small importance of the number of pairs in the audio captioning learning process in Appendix A. Our study compares the alignment variants considered in our zero-shot audio captioning framework against the current state-of-the-art methods, which employ audio encoders trained in either a supervised or an unsupervised manner. Our results indicate that the alignment through contrastive learning without audiovisual distillation does not fully match distributions, compared to DALI. Indeed, even without audiovisual distillation, $\text{DALI}_{\text{MMD}}$ and $\text{DALI}_{\text{OT}}^{\text{Att}}$ significantly outperform Shaharabany et al. (32), this system being our closest competitor, as it relies on a backbone trained in an unsupervised fashion.

Without audiovisual distillation, $\text{DALI}_{\text{MMD}}$ performs comparably to $\text{DALI}_{\text{OT}}^{\text{Att}}$, however, the latter improves with audiovisual distillation. We hypothesize that this occurs because $\text{DALI}_{\text{MMD}}$ is trained to learn the complete image token distribution, while $\text{DALI}_{\text{OT}}^{\text{Att}}$ only learns a part of it (due to the attentive optimal transport mechanism). Therefore $\text{DALI}_{\text{OT}}^{\text{Att}}$ features can take more advantage of the combination with image tokens. We verify qualitatively this hypothesis in the Discussion paragraph below. Interestingly, both $\text{DALI}_{\text{OT}}$ and the contrastive backbone benefit from the audiovisual distillation. Given the relatively low-performance scores prior to the distillation process, it is plausible that they primarily learn from the image captioner (as the audiovisual pseudo captions tend to rely solely on visual features due to the inadequate audio representations) which is shown to be already performing well for audio captioning (first line of results). We hypothesize that the second stage is effective even for poorly aligned backbones because the distillation occurs on a subset of AudioSet where the images align well with the audio. If this distillation were applied to the entire dataset, the performance of these backbones might decline substantially. These hypotheses will require further investigation in future work. Lastly, it is important to note that feeding the image tokens in addition to the $\text{DALI}_{\text{OT}}^{\text{Att}}$ ones, increases even more the performances, leaving the door open for further improvements.

**Discussion** Unlike standard contrastive methods, we do not average the output of the backbone, which allows us to employ token-level distribution alignment methods that do not rely on negative samples. Both Optimal Transport (OT) and Maximum Mean Discrepancy (MMD) focus on directly aligning the embedding's distributions. OT minimizes the cost of transporting one distribution to another, aligning them without the need for negative samples, and avoiding the issue of pushing different modalities into separate subspaces. Similarly, MMD measures the distance between kernel-projected mean embeddings, aligning the distributions without negative sampling. These methods promote a more cohesive embedding space, closely aligning different modalities and thus avoiding the modality gap inherent in contrastive learning approaches.

Indeed, Figure 4 shows a Principal Component Analysis (PCA) of the global representation (the average of the output tokens) of audio and im-

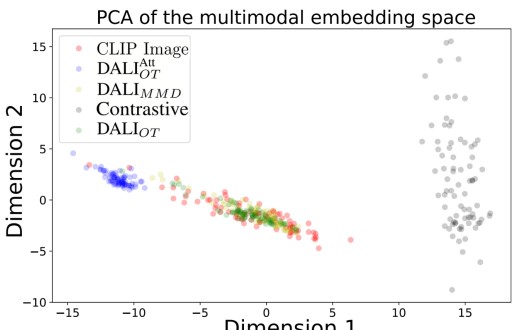

Figure 4: AudioCaps average tokens distribution. While contrastive learning maps the audio in a space separate from the image ones, MMD and optimal transport project in the same part of the space. The model trained using attentive optimal transport projects the audios in a space closer to the image, with marginal overlap.

ages from various backbones on a subset of the AudioCaps test set. In our audiovisual experiment, we observe a phenomenon akin to the one demonstrated by Liang et al.(15) for image-text contrastive learning. Specifically, models encode each modality in a narrow cone, resulting in the so-called modality gap where the embeddings of each encoder are confined to a subregion of the embedding space. However, our backbones, which are trained using distribution alignment, do not exhibit this issue. The embeddings generated by both the $DALI_{MMD}$ and $DALI_{OT}$ encoders closely resemble those of the image modality. In contrast, the $DALI_{OT}^{Att}$ ones, while located in the same region of the embedding space, remain distinct from the image tokens and exhibit minimal overlap. This indicates $DALI_{OT}^{Att}$ and image tokens are complementary and provides an explanation for why the former benefits more from audiovisual distillation than the $DALI_{MMD}$ model.

The root cause of this discrepancy lies in the training process: the cross-attention-based $DALI_{OT}^{Att}$ model is trained using only to the matching audiovisual tokens, whereas $DALI_{MMD}$ model is trained on the complete image distribution, which inherently includes a degree of image bias.

To verify this claim, we examined whether the aligned backbone with $DALI_{MMD}$ (after Stage 1) exhibits image bias. We prompted the captioner with "What can you see? Answer concisely" along with the audio tokens without the tuned prefix to avoid constraining it to generate only audio-related captions. Table 2 presents multiple captions generated using the $DALI_{OT}^{Att}$ and $DALI_{MMD}$ backbones. As shown, the encoder trained with MMD often produces captions with notable visual biases, which are absent in the outputs of $DALI_{OT}^{Att}$. These results confirm our hypothesis and align with the visualizations in Figure 4. Note that the captions may seem incomplete as we ask the model to answer concisely, focusing only on the main element of the scene . Complete generated captions by different models are provided in Appendix D.

| Ground Truth | $DALI_{MMD}$ | $DALI_{OT}^{Att}$ |
|---|---|---|
| A woman is speaking from a microphone | A woman in a black shirt and a red scarf is speaking into a microphone. | A woman |
| A male voice and a machine buzzing | A man is using a drill to make a hole in a piece of wood. | A person's hand holding a tool. |
| A bell is ringing | A person and a dog standing in front of a large machine. | Bell |
| A long burp ends in a sigh | A man with a beard and mustache. | A man |
| A chainsaw cutting as wood is cracking | A man cutting a tree trunk with a chainsaw. | A lawn mower |
| A man speaking as rain lightly falls followed by thunder | A man wearing a black shirt and a grey suit. | A man |
| A vehicle engine revving and squealing tires | A car racing on a track. | A car. |

Table 2: Ground-truth audio captions, and captions generated by our audio models (after stage 1, without prefix tokens) by asking "What can you see answer concisely".

**Assessing generalization ability** The Clotho dataset presents a greater challenge than AudioCaps, primarily due to the complexity of the captions, which is reflected in the relatively lower performance of all the methods. The audio tracks in AudioCaps are sourced from AudioSet, thus the distribution of audio events resembles that of the data on which our models are trained. In contrast, Clotho's audios

| | Alignment | METEOR↑ | ROUGE-L↑ | SPIDEr↑ |
|---|---|---|---|---|
| Ours | $\text{DALI}_{\text{MMD}}$ | **0.1067** | **0.2993** | 0.0655 |
| | $\text{DALI}_{\text{OT}}^{\text{Att}}$ | 0.1008 | 0.2850 | 0.0625 |
| | Contrastive | 0.0918 | 0.2490 | 0.0620 |
| | $\text{DALI}_{\text{OT}}$ | 0.1049 | 0.2765 | 0.0574 |
| Salewski et al. | CLAP* | 0.094 | 0.254 | **0.097** |
| Ablation of of audiovisual distillation (Stage 2-d) | $\text{DALI}_{\text{MMD}}$ | 0.1058 | 0.2768 | 0.0640 |
| | Contrastive | 0.0630 | 0.1777 | 0.0377 |
| | $\text{DALI}_{\text{OT}}^{\text{Att}}$ | 0.0666 | 0.2101 | 0.0355 |
| | $\text{DALI}_{\text{OT}}$ | 0.0617 | 0.1825 | 0.0299 |

Table 3: Clotho audio captioning performance. Similarly to AudioCaps, $\text{DALI}_{\text{OT}}^{\text{Att}}$ is performing, however, $\text{DALI}_{\text{MMD}}$ gives slightly better results. The bias learned by matching the complete image distribution seems to be beneficial for out-of-domain samples. (*): Trained in a supervised fashion using audio-caption pairs.

are sourced from FreeSound, a dataset distinct from AudioSet. Consequently, our model, having been trained solely on AudioSet's audio excerpts and images, may encounter unfamiliar concepts in Clotho, thereby testing its capacity for generalization. Table 6 presents the models' performance on Clotho's test set. The observed trends are consistent with those on AudioCaps where $\text{DALI}_{\text{MMD}}$ does not benefit from audiovisual distillation, while $\text{DALI}_{\text{OT}}^{\text{Att}}$ does. However, the absolute results differ, as $\text{DALI}_{\text{MMD}}$ marginally outperforms $\text{DALI}_{\text{OT}}^{\text{Att}}$ even after the second training stage. We posit that the image bias inherent in $\text{DALI}_{\text{MMD}}$ becomes beneficial when confronted with out-of-distribution data. Indeed what may be regarded as out-of-distribution for the audio backbone might have been seen by the image one (due to its bigger training set). Consequently, the learning of the image backbone's bias could potentially be beneficial. Notably, our method outperforms Saleswki et al., despite the fact that the latter employs a backbone trained using millions of audio-text pairs (CLAP). It is important to note that our method differs from Saleswki et al. in that we perform unsupervised instead of supervised zero-shot audio captioning.

## 6 Conclusion

We presented a novel methodology for unsupervised zero-shot audio captioning, effectively leveraging advanced image captioning models, such as Llava. We adapted it to the audio captioning task by making use of prefix tuning and using innovative token distribution alignment techniques (using MMD or optimal transport with a cross-attention mechanism) that successfully bridged the modality gap between audio and visual inputs. Our comprehensive evaluation, both quantitative and qualitative, demonstrates that our method achieves state-of-the-art results in unsupervised zero-shot audio captioning. Remarkably, our approach, not using any audio recordings, not only matches but even sometimes, exceeds the performance of existing methods that rely on a backbone trained with extensive audio-text data such as CLAP. Moreover, our method requires only raw videos without any audio annotations, significantly enhancing its potential scalability. This positions our approach as a promising direction as a leading direction for the future of audio captioning.

**Limitations** Although capable of addressing partial audiovisual mismatches, our method remains insufficiently robust to complete cross-modal mismatches, such as instances where the image is entirely unrelated to the audio. We reserve addressing this limitation for future work. Moreover, to achieve even higher performance in audio captioning, our method would benefit from a degree of supervision. The requirement for an external supervisory source (*e.g.*, audio-text pairs) arises due to the method's dependence on audiovisual alignments, which makes it challenging to learn certain sounds because they cannot be seen. For example, it is almost impossible to generate a caption for the sound of the wind by merely using images.

**Acknowledgement** This work was supported by the Audible project, funded by French BPI. This work was performed using AI resources from GENCI-IDRIS (Grant 2023-AD011014885).

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

The appendix is organized as follows: The first part presents additional experiments that aid in understanding the behavior of our encoders, followed by a detailed explanation of the pre-filtering process performed on AudioSet and the training parameter details.

# A    Impact of the number of image captions

To determine whether the tuned prefixes only specify the task and help avoid visual artifacts in the captions, we analyzed the performance of the method by varying the number of image- audio caption pairs. Figure 5 illustrates that only 16 captions are required to achieve good performance, and increasing this number does not lead to further improvement. In fact, while the SPIDER, CIDEr, and ROUGE metrics remain stable, the SPICE, METEOR, and BLEU scores even show a slight decline. This suggests that prefix tokens only serve to specify the task to the language model and do not contribute to learning the captioning task itself. We hypothesize that this prefix tuning could potentially be replaced by a carefully chosen hand-crafted prompt.

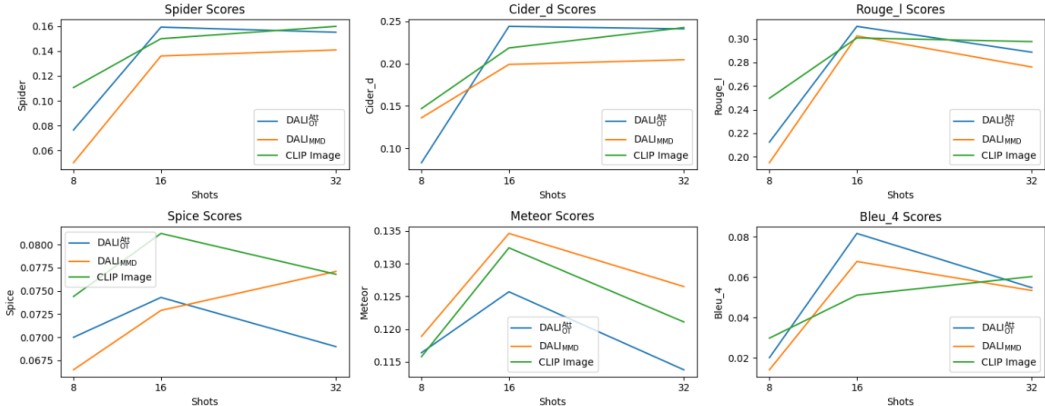

Figure 5: Captioning performances according to the number of image captions. After 16, the performance does not improve, indicating that the prefix tuning does not play an important role in the learning process, it just specifies the task.

# B    Deeper analysis of the learned distributions

Figure 6 presents a 2-D projection through PCA of CLIP image tokens along with the audio tokens learned by $DALI_{OT}^{Att}$, $DALI_{MMD}$, and contrastive learning for the audiovisual pair shown in the figure ("A sound of a guitar playing"). The contrastive learning method maps the audio tokens to a very restricted subset of the space, without any overlapping with the image distribution. Interestingly, while $DALI_{MMD}$ only learns to align the expected value of the distribution, it appears to fit the full image distribution quite well. $DALI_{OT}^{Att}$ exhibits similar behavior but is less noisy: the tokens are either very close to the image tokens or much further away. This behavior is inherited from the attentive optimal transport, which assigns low weights to points considered as mismatches.

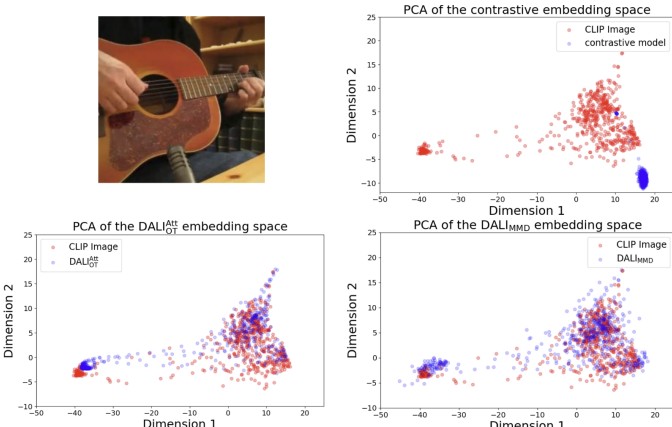

Figure 6: Audio and image tokens distribution. While contrastive learning encodes tokens in a completely separated subspace, the MMD learns a distribution similar to the image one, noisier. The attentive optimal transport fits more the image distribution, except in some points that are mapped further.

Figure 7 illustrates the optimal transport weights computed by the cross-attention mechanism for the given image and its associated audio (an emergency vehicle siren). The size of the circles represents the weights. Although the overall distribution appears uniform, a closer inspection of the bottom right section reveals that many tokens have scores close to zero. These tokens likely represent the road, which does not produce any sound and thus has no corresponding match in the audio tokens. Consequently, these tokens are not considered in the calculation of the transportation coupling

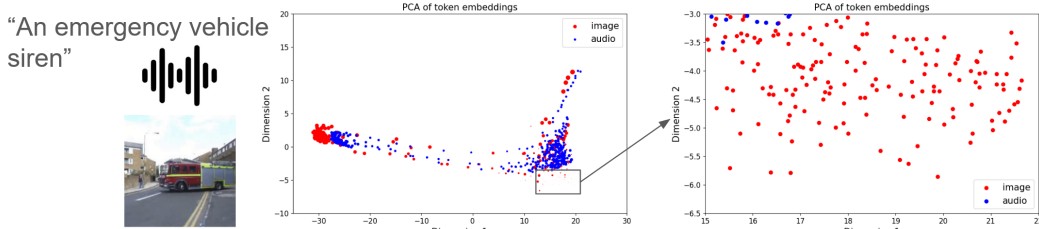

Figure 7: DALI$_{OT}^{Att}$ Cross-Attention scores. The size of the dots represents the weights of the transport defined by the cross-attentions. The image and its associated audio show a partial mismatch: the audio only contains the siren sound and the image also shows the road. All the tokens with low weights belong to the same part of the space which might indicate that they represent similar information such as the road.

## C   AudioSet pre-filtering to limit audiovisual discrepancies

As AudioSet primarily focuses on audio rather than audiovisual content, it features audio events that may be unrelated to the visual event depicted. For example, many videos feature music as an audio event while displaying a static image of the album cover.

To address this issue, we filtered out audiovisual discrepancies. To achieve this, we generated captions for the first frame of each second of a video using BLIP2, which we then encoded into GPT2-embedding space. Subsequently, we computed the distance between the embedding of each frame's caption and the embedding of the corresponding AudioSet labels. This process is illustrated in Figure 8. The average distance across the video frames was used as the distance between the video and the audio. We retained the 500,000 pairs with the smallest distance.

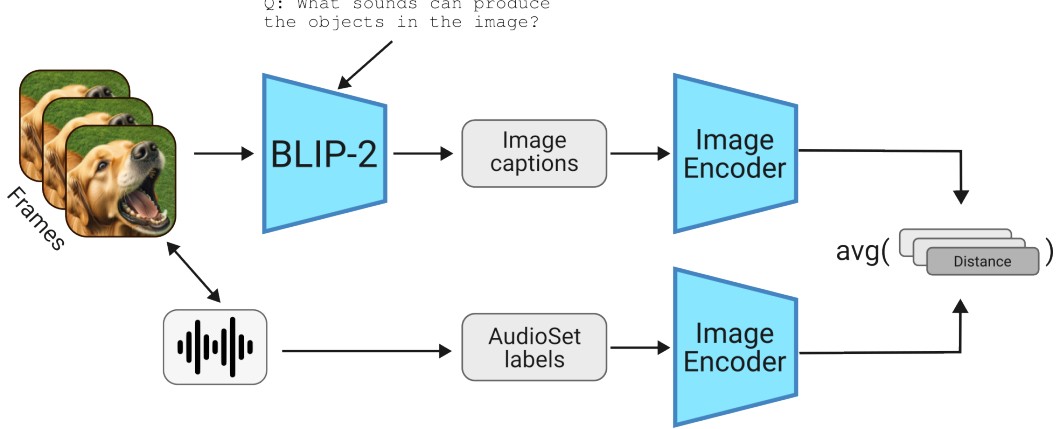

Figure 8: Discreptancies filtering process: 10 frames of the video are captioned by BLIP2, captions are embedded in a text space and compared to the embedding of the class labels. The average of the distances of the frame is considered as the distance between audio and video.

## D    Caption examples

| Ground Truth | Contrastive | DALI$_{OT}^{Att}$ |
|---|---|---|
| Humming of an engine with a woman and men speaking | A man with a vanity | A car engine |
| Rustling with some distant banging and people speaking in the distance | A loud noise | A person is talking |
| Wind is blowing and heavy rain is falling and splashing | Rain | The sound of a waterfall |
| A woman is speaking from a microphone | Describe the sound that can be heard in this scene. | A woman speaking |
| A bell is ringing | The sound of a large stone structure. | The sound of a bell ringing |
| A man speaking as rain lightly falls followed by thunder | The sound of a building being built | A man talking |
| Man speaking and clinking dishes | A person is seen eating a donut | A person is talking |
| A chainsaw cutting as wood is cracking | The sound of chainsaws and dirt being moved. | The sound of a chainsaw cutting through a tree. |
| A woman speaks and a cat meows | '[0.0]' | A person is talking |
| A male voice and a machine buzzing | The sound of a metal being cut or a saw | A person is talking |

Table 4: Example of captions from our method after stage 1, using contrastive backbone and DALI$_{OT}^{Att}$

Table 4 shows multiple examples of captions generated by the model using the contrastive backbone and DALI$_{OT}^{Att}$, both after the first stage of training. While the captions from the latter are accurate, using the contrastive backbone generates quite noisy/bad captions. This outlines the fact that contrastive learning is not well suited to swap the encoder of a modality with the one from another.

## E    Training details

For the MMD alignment, following (54) we use a mixture of $K$ kernels spanning multiple ranges (being the average pairwise squared distance between samples multiplied by $k$ with $k = \{0, ..., K\}$), with $K = 5$. As far as optimal transport is concerned, the first epoch of the first stage is performed with uniform transport weights. A decreasing value of $\lambda$ is then used: from 500 to 100 at the end of the 5th epoch. $\lambda = 100$ is then used for the remaining epochs. We observed that the training is quite robust to small variations of $\lambda$, as long as the first epoch is performed with uniform weights. We used the Python Optimal Transport (POT)(55) library to compute efficiently the EMD. In the prefix tuning stage, we learned 16 prefix tokens on different numbers of image and audio caption pairs. We used cosine decay and a learning rate of 2e-4 for both the prefix tuning and the MLP tuning in the second stage. The prompt used for all the experiment is: "What sound can be heard in this scene ?". We used version 1.5 of Llava: the image encoder being the ViT-L@336px version of CLIP (14), and the mapping network between image tokens and the LLM embedding space a 2 layer-perceptron. As for the audio encoder, we start from a pre-trained CAV-MAE (36) model in order to speed-up convergence. This model consists of 12 transformer encoder layers of dimension 768 and one final linear layer that maps the output of the transformers to a 1024-dimensional space (dimension of the CLIP encoder). Note that this encoder is way smaller than the image one (88M vs 428M parameters),

because the training set used is 2 orders of magnitude smaller than the CLIP one.
All the trainings were performed using 4 Nvidia A100 gpus, and all the steps of the training can be performed within 30 hours of compute on those machines.

## F  Attentive Optimal Transport details

For $DALI_{OT}^{Att}$, we computed the weights of the optimal transport by averaging on the feature dimension the tokens resulting from the cross-attention and then feeding them through a softmax with temperature (we used $\tau = 20$).

More formally, the weights of the transports $\alpha^{Att}$ and $\beta^{Att}$ are computed as follow:

$$\alpha^{Att} = \text{softmax}(\text{softmax}(\frac{(W_a^Q A)(W_i^K I)^T}{\sqrt{d}})W_i^V I/\tau), \tag{6}$$

$$\beta^{Att} = \text{softmax}(\text{softmax}(\frac{(W_i^Q I)(W_a^K A)^T}{\sqrt{d}})W_a^V A/\tau), \tag{7}$$

with $A$ and $I$ being the audio and image tokens, $W_i^Q, W_i^K, W_i^V$; $W_a^Q, W_a^K, W_a^V$ the image and audio projections, respectively, and $\tau$ a hyper-parameter.

## G  Additional captioning metrics

| | Alignment | BLEU$_4$↑ | METEOR↑ | ROUGE-L↑ | CIDEr↑ | SPICE↑ | SPIDEr↑ |
|---|---|---|---|---|---|---|---|
| | Image only* | 0.0511 | 0.1324 | 0.3008 | 0.2186 | 0.0812 | 0.1499 |
| Ours | $DALI_{OT}^{Att}$ | **0.0817** | 0.1277 | 0.3106 | 0.2441 | 0.0743 | 0.1592 |
| | $DALI_{MMD}$ | 0.0678 | **0.1346** | 0.3025 | 0.1991 | 0.0729 | 0.1360 |
| | $DALI_{OT}$ | 0.0461 | 0.1332 | 0.2923 | 0.1979 | **0.0866** | 0.1422 |
| | Contrastive | 0.0590 | 0.1215 | 0.2914 | 0.2290 | 0.0779 | 0.1524 |
| | $DALI_{OT}^{Att}$+Image | 0.0677 | 0.1257 | 0.3061 | **0.3063** | 0.0828 | **0.1946** |
| Shaharabany et al. | ImageBind | n.a | 0.086 | 0.082 | 0.092 | n.a | n.a |
| Salewski et al. | CLAP** | 0.068 | 0.123 | **0.331** | 0.281 | 0.086 | 0.183 |
| Ablation of audiovisual distillation | $DALI_{OT}^{Att}$ | 0.0523 | 0.11 | 0.2901 | 0.2227 | 0.0659 | 0.1443 |
| | $DALI_{MMD}$ | 0.0694 | 0.1330 | 0.3018 | 0.1976 | 0.0794 | 0.1385 |
| | $DALI_{OT}$ | 0.0476 | 0.1062 | 0.2709 | 0.1769 | 0.0597 | 0.1183 |
| | Contrastive | 0.0268 | 0.0728 | 0.1838 | 0.1287 | 0.0455 | 0.0871 |

Table 5: AudioCaps captioning results using 16 image captions. (*): No alignment. (**): Trained in a supervised fashion using audio-caption pairs.

| | Alignment | BLEU$_4$↑ | METEOR↑ | ROUGE-L↑ | CIDEr↑ | SPICE↑ | SPIDEr↑ |
|---|---|---|---|---|---|---|---|
| Ours | $DALI_{OT}^{Att}$ | 0.0583 | 0.1008 | 0.2850 | 0.0685 | 0.0565 | 0.0625 |
| | $DALI_{MMD}$ | **0.0587** | **0.1067** | **0.2993** | 0.0713 | 0.0598 | 0.0655 |
| | $DALI_{OT}$ | 0.0396 | 0.1049 | 0.2765 | 0.0567 | 0.0582 | 0.0574 |
| | Contrastive | 0.0358 | 0.0918 | 0.2490 | 0.0682 | 0.0562 | 0.0620 |
| Salewski et al. | CLAP* | 0.029 | 0.094 | 0.254 | **0.140** | 0.053 | **0.097** |
| Ablation of of audiovisual distillation | $DALI_{OT}^{Att}$ | 0.0193 | 0.0666 | 0.2101 | 0.0408 | 0.0302 | 0.0355 |
| | $DALI_{MMD}$ | 0.0483 | 0.1058 | 0.2768 | 0.0659 | **0.0621** | 0.0640 |
| | $DALI_{OT}$ | 0.0109 | 0.0617 | 0.1825 | 0.0321 | 0.0278 | 0.0299 |
| | Contrastive | 0.0167 | 0.0630 | 0.1777 | 0.0460 | 0.0293 | 0.0377 |

Table 6: Clotho audio captioning performance. Similarly to AudioCaps, $DALI_{OT}^{Att}$ is performing, however, $DALI_{MMD}$ gives slightly better results. The bias learned by matching the complete image distribution seems to be beneficial for out-of-domain samples. (*): Trained in a supervised fashion using audio-caption pairs.

