# OpenReview forum: "An eye for an ear: zero-shot audio description leveraging an image captioner with audio-visual token distribution matching"
_NeurIPS.cc/2024/Conference — NeurIPS 2024 poster_

### Official Review · Reviewer_aH67 · 2024-06-15

**Soundness:** 2
**Presentation:** 1
**Contribution:** 2
**Rating:** 3
**Confidence:** 4

**Summary:**

This paper proposes to use a well-trained visual LLM (specifically Llava-v1.5) to perform audio captioning in a zero-shot fashion. The authors propose a training framework which comprises 2 stages and 5 sub-steps. The authors also propose to use Maximum Mean Discrepancy (MMD) and optimal transport (OT) as the loss function to align audio and visual representation space.

**Strengths:**

1. The two loss functions are novel directions to explore for alignments between different modalities
2. This paper explores audio-visual LLM application scenarios which is timely.

**Weaknesses:**

1. The motivation is not clear. The author must specify in which scenario zero-shot audio captioning is needed, with concrete examples that the proposed method will help in practice, such as generating a particular captioning style or targeting a specific type of audio. As far as I can see in the paper, the authors only did experiments on two standard tasks (AudioCaps and Clotho) and both tasks have a reasonably-labelled size, and the performance is far from the state-of-the-art.

2. The method lacks theoretical grounding or insights. The author says they randomly choose one frame from the 10 frames in a video as the image and train to pull the image representation space to be close to the audio space. The author did not analyse how related the image is to the audio. How often does the image reflects what can be heard, and how often it does not?

3. The experimental setup is questionable.
  - The author says clearly in line 167 that they performed p-tuning with audio captioning data "We trained the additional tokens using image-text pairs, with the text describing the sounds associated with the images". This clearly sets this method apart from "zero-shot learning" which refers to having no data at all. If the author means "zero-shot" for a specific audio caption dataset, they need to be clear about it.
  - Regarding experimental results: I wonder if the comparison between the proposed method and ImageBind is fair. They also did not say how much data they used for ImageBind p-tuning. If the authors just took the numbers from Shaharabany et al., then putting the numbers in the same table is clearly unfair. I might have missed something here but without a convincing explanation, I do not think the experiments are valid. Why is ImageBind not included in Table 2?
  - I would be interested to see whether adding audio to the audio-visual description would give better performance or not. It seems that the audio information is all included in the visual information. Getting back to Weakness 1 - what is the point of doing this task then? What extra did you gain?

A few writing problems:
1. The referencing style makes it difficult to locate various papers.
2. Fig. 2 captions are incomplete.

**Questions:**

See Weaknesses.

**Limitations:**

There is a limitations section but no discussion on potential negative social impact.

---

> ### Author Rebuttal · Authors · 2024-08-07
>
> Thank you for your review. We’re glad to hear that you found the exploration of our **loss functions as novel directions for alignments between different modalities** and the **timely application scenarios of audio-visual LLM** valuable. Below, please find a point-by-point response to your feedback.
>
> >The motivation is unclear. The author should specify practical scenarios where zero-shot audio captioning is needed.
>
> We posit that Zero-shot audio captioning is particularly valuable in scenarios where annotated audio data is scarce or unavailable. Current audio captioners like Audio Flamingo and Pengi, trained on nearly all publicly available audio-text pairs, seem to have reached the limits of the supervised approach. In contrast, the abundance of videos makes unsupervised audiovisual training a viable way to scale further.
> For example, domain-specific topics (like wildlife sounds) that lack supervised datasets but non-annotated videos are available, which is realistic given that there exist only two "real" academic audio captioning datasets.
>
>
> >The author randomly selects one frame from a video without analyzing the relevance of the image to the audio.
>
> We indeed chose one random frame from the video to match its distribution with the audio and analyzed the image-audio relatedness. The paragraph Datasets of Section 4 as well as the full section C of the Appendix details this process: (line 579-581) “As AudioSet (...) features audio events that may be unrelated to the visual event depicted (...) we filtered out audiovisual discrepancies”. Pairs were chosen “by computing the similarity between the noisy audio labels and an image caption generated by BLIP-2” (line 269-270). This filtering considers the entire video,averaging similarity scores across frames sampled at one-second intervals.
> To address how often the image reflects the audio, we retained the 500k cleanest pairs, which best retain related pairs (line 269: “we train our models on a subset of 500k videos from AudioSet”). Additionally, since the image and audio can contain partial, non-common information, we used attentive optimal transport to learn only the common parts of the distributions (Section 3.2, paragraph 3).
>
>
> >The author states that they used image-text pairs for p-tuning, which differs from "zero-shot learning"
>
> We acknowledge your concern about potentially breaking the “zero shot” set-up with our handling of data. To clarify this concern, as we explained in line 53 of the submission, and consistent with Salewski et al., we define zero-shot audio captioning as **the model's ability to generate audio captions without training with manually assembled audio-caption pairs**, as they also used textual-only descriptions (AudioSet tags from a "keyword bank") as opposed to actual audio files and their corresponding captions.
>
> While we did use captions for prefix tuning, we emphasize that we **never** used audio-caption pairs for the targeted audio captioning task, which would indeed compromise the experimental setup. Instead, we used a few image-caption pairs without accessing the corresponding audio files. This use was solely to retask the LLM to create audio-centric captions.
> We maintain that our approach effectively addresses the zero-shot audio captioning task, allowing direct comparison with Salewski et al. and other methods.
>
>
> >Is the comparison with ImageBind fair? Data used for ImageBind p-tuning isn't specified. Why is ImageBind absent from Table 2?
>
> Firstly, we would like to clarify that our work does not compare directly to ImageBind, as it lacks the capability to perform audio captioning. Instead, it is a model that maps multiple modalities to a shared space. Hence, comparing our method directly to ImageBind is not feasible.
>
> We instead compare with Shaharabany et al., which, as described in Lines 108-114. This work is the closest to ours in terms of backbone training setup, as it has been trained using audio-image and image-text pairs but **not** audio-text pairs.
>
> Second, Shaharabany et al. did not perform p-tuning, relying instead on an “audibility score” (part of the training loss, core of their work). In our case, we used p-tuning for the same purpose (using only image-text pairs). Since neither work used audio-text pairs, comparison is feasible and fair.
>
> We acknowledge that, as ImageBind also contains an image encoder, prefix tuning can be applied. Unfortunately, as Shaharabany et al.’s code is not available, we couldn’t test his interesting idea.
>
> Finally, the work of Shaharabany et al. is not included in Table 2 as they did not report their results on Clotho. We will clarify this by extending Table 2’s caption to ”Shaharabany et al. did not report results on Clotho.”
>
> >Does adding audio to the audio-visual description improves performance ? What was the purpose of this task and what additional value did it provide?
>
> We would like to recall, that our method is designed to handle multiple modalities, such as image, audio AND audio-visual for the zero-shot audio captioning task. This is achieved by substituting or concatenating modality-specific tokens.
> As demonstrated in Table 1, our method performs better when both image and audio are used together (6th row, labeled DALI_OT^att+Image) compared to using the image alone (1st row). This highlights the usefulness of including audio, offering details about sound sources occluded or out of view in the image, e.g., a ringing phone inside a bag.
>
> Our results show that using both modalities leads to more informative audio captions. We also show that using audiovisual captions as pseudo-captions to supervise an audio-only model in a second stage (audiovisual distillation), yields a more robust audio captioner. One core motivation was to leverage a powerful image captioner to enhance audio captioning, creating a virtuous cycle where both modalities mutually benefit. Thus, including audio proves significant value beyond what visual data alone can offer.

---

> > ### Comment · Reviewer_aH67 · 2024-08-12
> > **Response to the Authors**
> >
> > I thank the authors for providing detailed responses. However, I do not think my concerns are adequately addressed for the following reasons:
> > 1. In response to my point 1, the author did not provide examples where zero-shot audio captioning is useful, e.g. categories held out of training so that they could be evaluated in zero-shot category generalization, as pointed out by reviewer rTvm. I still doubt whether this task under the author's specific setup is useful or not.
> >
> > 2. In response to my question about the experimental setup:
> >   - I disagree with the authors (including the cited paper) that having audio captioning (description particularly used for audio) is a practical setup. After all, how would one generate an audio description when looking at an image? I do not see the point of the experiment when you have images and descriptions for audio for training. Why would we have such a weird dataset? This is fundamentally questionable from a practical point of view. To validate this point, please assume there are also no pairs of image and audio-captioning text and re-run the experiments.
> >   - In response to my point on comparison to "ImageBind", the authors clearly say "Shaharabany et al. did not perform p-tuning", whereas, given my concerns about their p-tuning setup, I think the authors should compare their method to Shaharabany et al. without p-tuning.
> >
> > Given all the above, I am not convinced by the task setup and experiments performed. Therefore, I would keep my score as it is.

---

> > > ### Author Response · Authors · 2024-08-13
> > >
> > > Thank you for taking the time to review our rebuttal. Let us address the last concerns which are probably due to a misunderstanding.
> > >
> > > On the practical scenarios for zero-shot audio captioning, two points need further clarification here:
> > >  - “zero-shot audio captioning” refers to the scenario where the system is trained without having access to audio-text pairs. It  is useful in diverse use-cases, notably, audio description for the hearing impaired... where footage is largely available, but audio annotations are sparse.
> > >  - While it is standard to evaluate “zero-shot category generalization” as part of classification systems, it is unusual to do so in the captioning application. In general, one would have to first extract “categories” (classes) from the ground-truth captions, which is a problem per se. We agree that the idea is interesting but it is clearly beyond the scope of this work, where the definition of “zero-shot” is different: i.e., no audio-text pairs during training.
> > >
> > > On the validity of the experimental setup:
> > > > I disagree with the authors [...] After all, how would one generate an audio description when looking at an image?
> > >
> > > We must disagree with the reviewer. It is clearly valuable to use audio description from audiovisual data, since such description correlates to a great extent to both the image and audio modalities.  For example, in audiovisual data, a textual description like "a train passing by" not only describes the visual element of a train but also implicitly refers to the expected sounds of a train—such as the clickety-clack of wheels on tracks and the whistle blowing.  This approach allows the model to shift focus from descriptions with heavy visual cues to auditory cues, akin to style transfer. To mention a few other examples: think of an image of a person holding a phone with the mouth open (“person speaking on the phone”), or one of an inclined bell (“bell tolling”), or a person operating a jackhammer, etc. We'd like to invite the reviewer to do an image search with these descriptions to see the result...
> > >
> > > > I do not see the point [...]
> > >
> > > First, please note that this is merely a pre-processing (which only uses 16 image-audio description pairs) and not the central component of the system.
> > > In subsection "Prefix-tuning with image-caption pairs," we explained that LLVMs tend to generate descriptions heavily focused on visual features rather than auditory ones even when asked, "What sounds can the objects in this image generate?" This visual bias negatively impacts audio captioning metrics and calls for a solution to shift descriptions toward auditory content. Importantly, no large image-caption (audio-centric caption) dataset is required, as the audio-centric descriptions **can be easily composed by hand**, given the small number needed. As demonstrated in Appendix A, using more than 16 pairs offers no additional benefit. Without the p-tuning process that reorients the model's descriptive style, the generated captions would continue to exhibit a strong visual bias, undermining the benefits of the audiovisual distribution alignment. This is confirmed by our preliminary experiments.
> > >
> > > > In response to my point on comparison to "ImageBind" [...] I think the authors should compare their method to Shaharabany et al. without p-tuning.
> > >
> > > We disagree. While Shaharabany et al. rely on an "audibility score" as part of their training loss to steer their models toward generating audio-centric descriptions, our approach employs p-tuning for the same purpose. This difference in methodology is important, as p-tuning effectively reorients the model's descriptive style from visual to auditory, addressing the inherent visual bias of LLVMs. Without p-tuning, our model would continue to produce visually biased descriptions. Therefore, comparing our method to Shaharabany et al. without p-tuning would not only undermine the intent of our approach but also fail to provide a fair comparison, given that both methods are designed to achieve the same goal through different means.

---

> > > > ### Comment · Reviewer_aH67 · 2024-08-13
> > > > **Further Questions**
> > > >
> > > > Thank you for providing more details. Please find my following concerns:
> > > >
> > > > 1. In those examples given, it seems visual-LLM could generate similar captions and hence achieve similar results since those scores are really in the very low region to draw any useful results (e.g. a SPIDEr value of around 0.19 in comparison to the SOTA which is over 0.50). Looking at Table 1, the comparison to the image-only baseline is actually __mixed__. A significance test needs to be performed here to show real improvements.
> > > >
> > > > 2. I wonder whether the main gain comes from the p-tuning. Since the p-tuning requires some few-shot samples, and hence the selection of those samples is important and I am not sure if that should be counted as "zero-shot" or should be thought of as "few-shot". Given the above two, I strongly think that the authors should study what happens without p-tuning and what p-tuning brings to this training pipeline.
> > > >
> > > > 3. Regarding the task itself, it would be interesting to see if the model could capture "audio" that is not included in any image training data. The examples such as "a man speaking" must have appeared multiple times in the image-text training pairs.
> > > >
> > > > Overall, I am not sure if the results are strong enough and the task still does not feel practical to me. I have to keep my original rating.

---

> ### Author Response · Authors · 2024-08-13
> **Answer to further questions**
>
> Thanks for the last answer and those new remarks. However, we do not agree with your points:
>
> 1-Concerning the difference between image only and audio only, it's important to notice that our
> method targets AUDIO captioning, which consists in generating a caption while having access to an AUDIO and not an image. Therefore, there is no point in trying to compare audio and image performance.
>
> 2-The prefix tuning does not introduce any bias or unbalance between methods as the prefixes are ALSO used with the image-only method, because as previously mentioned, the visual-LLM off-the-shelf is not able to produce audio-centric captions.
>
> We would encourage the reviewer to perform a more careful reading of the paper as he/she seems to have missed a lot of important details which are very clearly stated.

---

### Official Review · Reviewer_rTvm · 2024-07-13

**Soundness:** 3
**Presentation:** 3
**Contribution:** 3
**Rating:** 6
**Confidence:** 4

**Summary:**

This paper presents a method to align tokens created by an audio encoder with those from an image encoder to allow zero-shot description of audio and audio-visual recordings. It introduces an attention-based matching to the alignment to be able to account for objects that appear in one modality but not the other (e.g., background music or the visual background). The paper compares two different alignment methods: maximum mean discrepancy and optimal transport, with the attention only applied in the case of optimal transport. And prefix tuning is used to guide the image captioner towards audio captions.

The model is trained on AudioSet and evaluated on AudioCaps and Clotho using standard captioning metrics. While several metrics are reported, the primary one used by the community is SPIDEr, and by this metric on AudioCaps (in-domain to training) the proposed approach when using audio-visual input (0.1946) outperforms a comparison supervised system (0.1830), an image-only system (0.1499), and other variants of the proposed system such as the audio-only version (0.1592). On the out of domain and audio-only Clotho dataset the supervised baseline performs best (0.097) followed by the MMD variant of the proposed system (0.0655). Note that state of the art performance on Clotho was around 0.3 for systems like QwenAudio and AudioFlamingo and around 0.5 on AudioCaps.

**Strengths:**

Significance:
* Cross-modal learning is an important problem, although it is not clear that zero-shot learning is necessary as there is a great deal of audio-visual content available, but if it is possible, then it seems worth exploiting.
* The ability to leverage visual captioning systems to bootstrap audio captioning systems is valuable, as the former have received much more attention than the latter.
* The issue of audio-visual correspondence is key in audio-visual processing and the proposed approach is an interesting way to deal with it.

Clarity:
* The paper is generally well written and easy to follow
* The diagrams are quite helpful in understanding the approach
* The paper includes a real discussion of its limitations
* The caption examples included in the appendices are quite informative. It would be nice if there were examples from all of the systems to compare

Technical correctness:
* The experiments are well organized and well conducted

**Weaknesses:**

Clarity:
* The use of many metrics dilutes the clarity of the results and their interpretation. The metrics should be prioritized and the systems they are comparing discussed accordingly. The discussion of results seems to treat all metrics as somewhat interchangeable and makes broad statements about which systems seem better than others. It would be much clearer and easier to follow if differences in performance on specific metrics were tied to qualitative differences in model behaviors and when there is one main metric (i.e., SPIDEr) to focus on that for overall system comparison.
* Line 96 states that "Audio data often lacks the richness and context provided through the visual modality, resulting in less precise and more subjective descriptions of the audio content." While I agree that audio captioning datasets typically contain more subjective descriptions of their content, I think this is an issue with the data that is selected to be captioned and the question asked of the captioner as opposed to a shortcoming of audio in general. With the right data and right task, audio captioning could be much better defined. I believe this sentence should be modified to focus on existing datasets as opposed to the general modality.
* No mention is made of the size of the various models or model pieces involved or how much data they were trained on. This is especially acute for comparing the proposed system to existing baselines from Shaharabany et al and Salewski et al.
* I can't understand what the ablations are showing, these should be described in the body of the paper, not just shown in tables.

Performance:
* Overall, the reported SPIDEr results (0.1-0.2) are fairly low compared to SOTA approaches for Clotho (0.3) and AudioCaps (0.5). It's clear that this zero-shot approach does not completely solve the problem. It would be useful to discuss this in the paper.
* Additionally, the differences in performance between the proposed variants are not that large, bringing into question the value of including multiple-such approaches.

Minor issues:
* Figure 2: sub-figures (d) and (e) are not mentioned in the caption
* Equation (4): please use \langle and \rangle as opposed to < and > for brackets
* Tables 1 and 2: Order the rows within each block intentionally. I would recommend ordering by SPIDEr.
* Figure 4: It's hard to distinguish between this many colors with this level of transparency. Maybe you could use different symbols instead of different colors.

**Questions:**

What exactly was done in the ablation studies listed in Tables 1 and 2? What do the rows mean? What was taken away?

Why wasn't the attention weighting used with MMD in addition to OT?

Does the audio-visual alignment treat each instance as a "bag of tokens"? If so, would there be value in considering the sequence of tokens in this alignment as opposed to just their distribution?

Were any categories held out of training so that they could be evaluated in zero-shot category generalization? Or was the focus only on zero-shot modality generalization?

Was the filtering of audiovisual discrepancies described in appendix C applied to both training and test data or just to training data?

**Limitations:**

The paper has a section focusing on limitations and does an excellent job discussing them, providing true limitations of the approach.

---

> ### Author Rebuttal · Authors · 2024-08-07
>
> Thank you for your insightful review. We’re happy to hear that you found our paper **well written and easy to follow**, appreciated the **helpful diagrams**, and recognized the **value in our approach to addressing audio-visual correspondence and leveraging visual captioning systems**. Below we address your concerns.
>
> > Using too many metrics muddies the results.
>
> We understand that using many metrics complicates the reading. Here's a simplified analysis:
> CIDEr emphasizes the precision of n-grams (targeting key words), while SPICE focuses more on the semantic content, making it complementary to CIDEr. SPIDEr, an average of both, is therefore used as the primary metric.
>
> On AudioCaps, all distribution alignment methods outperform the contrastive baseline (SPIDEr: 0.0871) after the first stage. While MMD (0.1385) performs better than standard OT (0.1183), attentive OT improves even further (0.1443).
> After distillation, all methods improve except for MMD (0.1360), with attentive OT remaining the best (0.1592) and the contrastive baseline improving significantly (0.1524). These results confirm the intuition that MMD learns a visual bias and therefore does not benefit from the distillation.
> Note that other metrics and especially in ROUGE_l, the contrastive approach remains less effective than other methods (0.2914 against 0.3025 and 0.3106 for MMD alignment and attentive optimal transport respectively).
>
> On Clotho, MMD’s visual bias is beneficial, outperforming other methods after the first stage (0.0640 vs. 0.0377, 0.0299, and 0.0355 for contrastive, OT, and attentive OT, respectively). Post-distillation, MMD does not improve (0.0655), while other methods do, with attentive OT (0.0625) and MMD remaining slightly ahead of the contrastive approach (0.0620). ROUGE_l shows similar trends to AudioCaps.
>
> This discussion aligns with the paper’s conclusions. We'll revise the manuscript accordingly.
>
>
> >Line 96 suggests audio data lacks richness. However, this is a data selection and task issue, not an audio modality shortcoming.
>
> We'll revise Line 96 for clarity: "Current audio captioning datasets often fall short in capturing the richness and context compared to what the vision datasets offer. This limitation arises from both the selection of data for captioning and the nature of the questions posed to captioners, leading to descriptions that are less precise and more subjective."
>
>
> >No mention is made of the size of the various models or
> how much data they were trained on.
>
> We agree mentioning this information is important. We will revise as follows:
>
> - Lines 104-107: “ZerAuCaps uses CLAP (an 80.8M-parameter model trained on 3.4M audio-text pairs) (...) to prompt an LLM (OPT 1.3B)”.
> - Lines 109-111: “The authors adopted ImageBind (using an audio encoder of 86M parameters trained on AudioSet) (...) They fine-tuned a GPT-2 model (117M parameters) (...)”
>
> > The reported are low compared to SOTA. What is the value of including multiple such approaches.
>
> We want to clarify that our approach extends image captioners to audio without compromising their original image performance. While better audio performance might be achievable by sacrificing some image performance (typically by unfreezing the LLM), our focus is on learning from videos for scalability due to their widespread availability. To our knowledge, this is the first attempt at this problem. This is a first step towards large-scale training, and a model trained on 500k videos cannot be compared with others trained on millions of audio-text pairs. Future work could scale this method further to compete with supervised approaches.
> For now, exploring alternative distribution alignment methods is valuable, but given the minor differences between methods, future work might focus on a single alignment method.
>
> >What was done in the ablation, what was taken away?
>
> Tables 1 and 2 present ablations for different training stages. The rows show model performance after the first stage, which includes only distribution alignment and prefix tuning. What was taken away is the role of the second stage: audiovisual distillation (as written in the Table captions: “Ablation of audiovisual distillation”). We will clarify this further in the text.
>
> >Why wasn't the attention weighting used with MMD in addition to OT?
>
> In attentive OT, cross-attention is applied in the token space to weight each sample of the empirical distributions. MMD, however, computes the expectation of the distribution in the kernel space, so weighting tokens before averaging would significantly alter the distance formulation, which is why we did not initially explore this.
> We acknowledge this is an interesting experiment and hace since tested it. The results, shown in Table B reveal that while performing well across all metrics,it underperforms in CIDE_r. The model generates overly detailed captions, which affects CIDE_r (measuring n_grams) but not in SPICE.
>
> >Does the audio-visual alignment treat each instance as a "bag of tokens"? Would there be value in considering the sequence of tokens ?
>
> Audio-visual alignment does treat each instance as a "bag of tokens". However, our preliminary experiments revealed that token order does impact llava's performance in audio captioning with images (on AudioCaps).
> To support this claim, we shuffled image tokens before feeding them to the LLM and found that the results were similar to those without shuffling (see Table A). This indicates that the sequential order of image tokens is not crucial for this task.
>
>
> >Were any categories held out of training to be evaluated in zero-shot category generalization?
>
> We only focused on zero-shot modality generalization, but exploring other aspects is worth pursuing in future work.
>
> >Was the filtering applied to both training and test data ?
>
> The filtering was applied only to the training set to keep test performance comparable with other methods. We will clarify this in Line 271.

---

> > ### Comment · Reviewer_rTvm · 2024-08-12
> > **Response to rebuttal**
> >
> > I would like to thank the authors for their rebuttal. It has resolved several of my questions and issues around clarity in the paper. Several issues remain, however.
> >
> > > Using too many metrics muddies the results.
> >
> > This summarization is useful and improves over what was in the manuscript original, although it is still not entirely easy to follow.
> >
> > > We want to clarify that our approach extends image captioners to audio without compromising their original image performance.
> >
> > Is this demonstrated quantitatively in the paper? If it could be, then that would go a long way to resolving this concern and would likely increase the significance of the work.
> >
> > As it currently stands, I have read the reviews and rebuttals and would still like to keep my rating as is.

---

> > > ### Author Response · Authors · 2024-08-13
> > >
> > > Thank you for your positive feedback.
> > >
> > > The original image captioner's performance remains **necessarily**  intact. Indeed, **all** the parameters of the original system, especially the LLM’s, are kept frozen during the whole process. Hence, merely  re-using the original image backbone (along with its MLP), we do recover the original model and indeed the original performance. Therefore, there is no need to perform an additional evaluation as the original model remains **intact**.
> > >
> > > We do hope this clarification resolves your last concern.

---

> > > > ### Comment · Reviewer_rTvm · 2024-08-13
> > > > **Response**
> > > >
> > > > Thank you for providing this additional clarification about the original visual model. Given that the proposed system does maintain its visual performance while also gaining the ability to perform audio-only captioning (on Clotho) and improving its performance for audio-visual captioning (AudioCaps), I do think the relatively poor absolute performance can be overlooked. I will raise my rating by one point to 6: Weak Accept.

---

### Official Review · Reviewer_7jnN · 2024-07-14

**Soundness:** 3
**Presentation:** 2
**Contribution:** 2
**Rating:** 6
**Confidence:** 3

**Summary:**

The paper proposes a method for how to adapt VLM to perform audio captioning. The pipeline is as follows: 1. perform few-shot prefix tuning on images to caption the audio. 2. Use multi-modal alignment methods, namely MMD or OT, to align token space distribution of audio and visual. 3. Distillation for audio-only model from the visually-informed audio caption model. Experiments show that the proposed method outperforms contrastive-based alignment and also the pretrained CLAP model that is used for audio captioning.

**Strengths:**

1. The main novelty of the paper lies in its usage of MMD / OT to perform multi-modal alignments in audio-visual space, though MMD / OT is not new and has been extensively explored in cross-domain matching literature.

2. The paper is well-written and easy to follow.

**Weaknesses:**

1. There is lack of theoretical derivation or enough insights for showing why the OT and MMD can outperform the standard contrastive learning. Figure 4 shows the PCA plots for different methods, however it is unclear why MMD or OT is able to stay closer to CLIP encoder. I think more explanations or theoretical justifications are needed.

2. The experiment results seem mixed, in particular Table 2 for Clotho. Contrastive method is better in several metrics without audio-visual distillation, in particular CIDEr and SPIDEr. Also the SPICE performance is very close to the proposed method. These two metrics, CIDEr and SPIDEr from my point of view is also very important since they are designed specifically for caption task. Why would Contrastive outperform the proposed methods in this dataset? But in AudioCaps, the proposed methods beat the contrastive method by a lot. The inconsistency results across these two datasets make me feel confused.

**Questions:**

See Weaknesees.

---

> ### Author Rebuttal · Authors · 2024-08-07
>
> Thank you for your review. We’re pleased to hear that you found our paper **well-written and easy to follow** and appreciated the **novelty in our use of MMD/OT for multi-modal alignment in the audio-visual space**. In the following, we'll address your comments in detail.
>
> >There is a lack of insights for showing why the OT and MMD can outperform, more explanations are needed.
>
> First, regarding the performance gap between our method and the contrastive baseline, we explain the need for distribution alignment to perform this task instead of contrastive learning in Section 3.2, lines 200-206: “Llava for instance, makes use of the full token distribution as input, naturally creating the need for a full token distribution alignment, to allow for swapping the pretrained encoder with a new one targeting a new modality. Moreover, contrastive learning faces the so-called “modality gap” problem where each modality is encoded in distinct sub-regions of the embedding space (see Figure 4). For all these reasons, replacing image tokens with audio tokens obtained by alignment through a contrastive learning approach may yield undesirable responses from the language model.”
>
> Concerning the modality gap that does not appear in our method, it is important to note that contrastive learning brings associated (positive) samples close together while pushing away unpaired (negative) samples. The cited paper [A] showed that by pushing away the negative samples in a high-dimensional space, combined with multiple false positive pairs, multimodal contrastive learning encodes each modality in a separate subspace, creating a modality gap. A more recent paper [B] goes further and shows that this phenomenon is purely due to the behavior of the contrastive loss itself in high dimensions and is not related to multimodal learning.
> Now, optimal transport and Maximum Mean Discrepancy (MMD) do not suffer from this issue, as they do not rely on negative samples but focus on explicit distribution alignment instead. However, their application in audio-visual alignment has been limited due to the common practice of averaging the final representation into a single token, which does not allow for the computation of distribution distances. Employing these losses for their ability to avoid the modality gap is a core motivation behind our work.
>
> We will modify the beginning of the Discussion paragraph in Section 5 to make clearer our motivations and why our methods do not suffer from such a modality gap. Specifically, we’ll complement it by adding the following paragraph:
> "Unlike standard contrastive methods, we do not average tokens, which allows us to employ sample-level distribution alignment methods that do not depend on negative samples. Indeed, both Optimal Transport (OT) and Maximum Mean Discrepancy (MMD) focus on aligning the distributions of the embeddings directly. OT minimizes the cost of transporting one distribution to another, effectively aligning the distributions in a way that does not rely on negative samples. This approach avoids the issue of pushing different modalities into separate subspaces. Similarly, MMD measures the distance between the kernel-projected mean embeddings of the distributions, ensuring that the overall distributions are aligned without the need for negative sampling. By leveraging these methods, we can achieve a more cohesive embedding space where different modalities are aligned more closely, thus avoiding the modality gap problem inherent in contrastive learning approaches."
>
> >Why would Contrastive outperform other methods in this Clotho without audiovisual distillation?
>
> First, on Clotho, before the audio-visual distillation, the alignment through MMD, which is also part of our method, yields significantly better results (sometimes twice as good) than the contrastive baseline (0.0659, 0.0621, 0.0640 against 0.0460, 0.0293, 0.0377 for CIDEr, SPICE and SPIDEr, respectively).
> As explained in the paper, the good performance of MMD is likely to be due to the learned image bias: line 343-344: “the image bias inherent in DALI_MMD becomes beneficial when confronted with out-of-distribution data.”
> It is important to note that Clotho is known to be a more challenging dataset than AudioCaps. Moreover, it contains new audio concepts not seen in AudioSet, representing an out-of-domain scenario for our models (trained only on AudioSet), hence the observed drop in performance, compared to AudioCaps.
> CIDEr and SPICE are indeed important metrics for captioning, and these metrics are roughly equivalent between contrastive and attentive OT (0.0302 0.0355 for attentive OT against 0.0293 0.0377 for contrastive in CIDEr and SPICE respectively). As both scores are fairly low, trying to interpret such small differences would be unreliable. However, those models exhibit significantly different performances when focusing on other metrics.
> In particular, the ROUGE_l, which measures the longest common subsequence, and takes the order of the words into account (interpreted as the “coherence” of the generated sentence), is significantly higher in our method (0.2101 for attentive OT against 0.177 for the contrastive). A tentative interpretation could be that, while both methods failed to generate accurate enough captions, the attentive OT still encodes the audio in a space that can be "comprehended" by the LLM as it stays in the same region as the image tokens, ending up in a more coherent caption, while the contrastive approach encodes something that cannot be interpreted by the LLM, leading to hallucinations or incoherent captions.
>
> [A] Weixin Liang and Yuhui Zhang and Yongchan Kwon and Serena Yeung and James Zou (2022) Mind the Gap: Understanding the Modality Gap in Multi-modal Contrastive Representation Learning
> [B] Abrar Fahim and Alex Murphy and Alona Fyshe (2024) It's Not a Modality Gap: Characterizing and Addressing the Contrastive Gap

---

> > ### Comment · Reviewer_7jnN · 2024-08-13
> >
> > Thank the authors for providing detailed response, and it basically solves my concerns about the paper. I would like to raise the score to weak acceptance considering the novelty part of the paper and these additional insights brought by the rebuttal response.

---

> > > ### Author Response · Authors · 2024-08-13
> > >
> > > Thank you for your positive feedback

---

### Official Review · Reviewer_mzTo · 2024-07-31

**Soundness:** 3
**Presentation:** 2
**Contribution:** 3
**Rating:** 6
**Confidence:** 3

**Summary:**

The paper under review looks at the problem of captioning of short audio-video clips, leveraging existing multimodal large-language-models. While many strong image captioners exist due to the abundance of image/caption pairs for training data, for clips of non-speech audio, the amount of data available are comparably much less. To build a performant captioner that can produce accurate captions for clips contained either of both audio and visual modalities, the paper proposes a framework that leverages LVLMs, adapts them to produce more audio-centric descriptions via prefix tuning, and most critically "aligns" the distributions of the audio and visual tokens to match making the tokenized output from the visual or audio encoder interchangeable. The main novelty is the use of Optimal Transport to align the visual and audio modalities and modifying the loss with cross attention terms to improve the alignment. While this DALI_OT^Att is not a large improvement over DALI_OT or DALI_MMD, the experimentation of these modality alignment methods and demonstrated improvement over a constrastive loss is a novel contribution. The analysis and plot of the multimodal embedding space of the various approach to align the A-V modalities further demonstrates the effectiveness. Overall the paper does demonstrate how LVLMs can be utilized to build a strong A-V captioning system.

**Strengths:**

Goal of the paper, contributions, execution are all clear. There are a good set of experiments, visualizations of the expected improvement in alignment of the visual and audio token distributions, and overall the application of MMD or OT learning improves the alignment as demonstrated by the experimental results over the dominant baseline approach of contrastive learning.

**Weaknesses:**

The paper has framed the problem of A/V captioning in a somewhat narrow way of leveraging LLVMs with the only comparison to a strong supervised baseline in CLAP while there are others such as GPTo, Gemini 2 or Llama 3.

Section 3.1 is rather confusing. It seems Figure 2 was revamped and the references to the figure in section 3.1 was not completely updated. Figure 2 should also include a description of steps 2d and 2e in the descriptive caption (the description of 1c needs updating too I believe. The reference on line 177 for 2-c should probably be 2-b; line 190 2-c should be 2-d; and on line 196 2-d changed to 2-e. Overall, it would be clearer to first indicate the stage and then explain it, e.g. "1-a) prefix tuning the prefix tokens" rather than the other way around. Also each of the bolded titles in section 3.1 should correspond with a stage in Figure 2 and include the stage number for better clarity.

In section 3.2, it would be helpful to the reader to provide concrete equations on how to compute the quantities and eqns 1-5. Having these in an appendix would be fine. While its understood that the code will eventually be released, having the equations that the code is intended to implement is always useful for clarity.

**Questions:**

Has there been any analysis done in comparison to other large, strong multimodal LLMs such as  GPT-4o, Gemini 2 or Llama 3?

**Limitations:**

Yes

---

> ### Author Rebuttal · Authors · 2024-08-07
>
> Thank you for your thorough review. We’re glad to hear that you found our paper’s **goal, contributions, and execution clear** and appreciated the **good set of experiments and visualizations,** along with the improvement in alignment demonstrated by our **application of MMD or OT learning**. We are now addressing your comments in detail.
>
> > Section 3.1 is rather confusing. It seems Figure 2 was revamped and the references to the figure in section 3.1 was not completely updated. Figure 2 should also include a description of steps 2d and 2e in the descriptive caption (the description of 1c needs updating too I believe. The reference on line 177 for 2-c should probably be 2-b; line 190 2-c should be 2-d; and on line 196 2-d changed to 2-e. Overall, it would be clearer to first indicate the stage and then explain it, e.g. "1-a) prefix tuning the prefix tokens" rather than the other way around. Also each of the bolded titles in section 3.1 should correspond with a stage in Figure 2 and include the stage number for better clarity.
>
> We thank the reviewer for pointing out these corrections. We will ensure to make the following changes:
> - Figure 2 caption: Full training pipeline: a prefix tuning is performed using a few image-captions pairs (1-a). In the meantime, the audio backbone is aligned with the image backbone (1-b) through distribution alignment.
>  Audio captioning can then be performed by switching the image backbone with the audio backbone while adding the prefix tokens (1-c).
> Visually-informed audio captions are then generated using both audio, image, and prefix tokens. The MLP that maps the audio encoder to the language model is fine-tuned using these pseudo captions (2-d).
> The final inference is performed by forwarding the output of the aligned audio backbone to the trained MLP to obtain the LLM input (2-e).
>
> - References to Figure 2: line 177 “We align the distributions (Figure 2:1-b)”; line 190 “This procedure is illustrated in Figure 2: 2-d”; line 196 “The resulting audio captioner is shown in Figure 2: 2-e”
>
> >In section 3.2, it would be helpful to the reader to provide concrete equations on how to compute the quantities and eqns 1-5. Having these in an appendix would be fine. While its understood that the code will eventually be released, having the equations that the code is intended to implement is always useful for clarity.
>
> We agree with those remarks. Since the details of the $\alpha$ and $\beta$ computation are already given in Appendix F, and since the OT does not have a closed-form solution, the title of Appendix F will be changed to “Implementation details” and the following precision will be added to ease the understanding of the practical computations of MMD:
>
> The MMD distance between the audio token distribution ($X$) and the image token distribution ($Y$) is computed as follows:
>
>  $\text{MMD}(X, Y) = \frac{1}{m^2} \sum_{i=1}^{m} \sum_{j=1}^{m} k(x_i, x_j) + \frac{1}{n^2} \sum_{i=1}^{n} \sum_{j=1}^{n} k(y_i, y_j) - \frac{2}{mn} \sum_{i=1}^{m} \sum_{j=1}^{n} k(x_i, y_j)$
> where $k$ is the Gaussian kernel.
>
> >**Question**: Has there been any analysis done in comparison to other large, strong multimodal LLMs such as GPT-4o, Gemini 2 or Llama 3?
>
> It is important to keep in mind that the purpose of our work is to extend vision captioners so they can **also** perform general audio captioning (i.e., describe an audio event—possibly happening outside of the field of view—, such as ‘a dog barking in the street' or ‘a siren heard at a distance').
> Large Multimodal models like the suggested ones are trained to perform various tasks, including video captioning. However, they largely rely on the visual modality for that task, and the use of the audio modality is restricted to the speech present in the video.
>
> Therefore, no fair comparison can be made with them as they are not trained to perform the same task as us (general audio captioning).

---

> > ### Author Response · Authors · 2024-08-14
> > **Rebuttal answer**
> >
> > Dear Reviewer,
> >
> > I hope this message finds you well. We wanted to kindly follow up to see if our responses have addressed your concerns regarding our work. As the deadline is approaching in an hour, we would greatly appreciate any feedback you can provide before the end of the discussion period.

---

### Author Rebuttal · Authors · 2024-08-07

We would like to start by thanking all the reviewers for the time they spent reading carefully our work and their valuable feedback that helped us improve the quality of the submission.

We would like to underline an important contribution of our work: we add audio capability to an LVLM while **keeping its vision-only performances intact**. This is important to keep in mind as this allows for the use of the same model for audio-only, vision-only or audio-visual tasks. This is highlighted in the 4th bullet point of our contributions, line 80-81: “Our method supports both audio and audiovisual inputs, extending the image captioner’s capabilities **without compromising its performance in image analysis tasks**.”

Below, we provide a detailed response to each reviewer's comments.

---

### Decision · Program_Chairs · 2024-09-25

**Decision:**

Accept (poster)

**Comment:**

This paper presents a method to align tokens created by an audio encoder with those from a pre-trained image encoder (Llava-1.5) to allow zero-shot description (captioning) of audio and audio-visual data. It introduces an attention-based distribution matching to the alignment to be able to account for content that appear in one modality but not the other (e.g., background music or the visual background), the so-called modality gap. The paper compares two different alignment methods: maximum mean discrepancy (MMD) and optimal transport (OT), with the attention only applied in the case of optimal transport, over a baseline approach of contrastive learning (widely used). Prefix tuning (p-tuning) is used to guide the image captioner towards audio captions (i.e. mention audio-relevant content), which is also a novel application of this method.

The model is trained on AudioSet and evaluated on AudioCaps and Clotho using standard captioning metrics. While several metrics are reported, the primary one used by the community is SPIDEr, and by this metric on AudioCaps (in-domain to training) the proposed approach when using audio-visual input (0.1946) outperforms a comparison supervised system (0.1830), an image-only system (0.1499), and other variants of the proposed system such as the audio-only version (0.1592). On the out of domain and audio-only Clotho dataset the supervised baseline performs best (0.097) followed by the MMD variant of the proposed system (0.0655). Note that state of the art performance on Clotho was around 0.3 for systems like QwenAudio and AudioFlamingo and around 0.5 on AudioCaps. The proposed model also maintains performance on the image task.

Reviewers agree that the paper is well written and that the goal of the paper, its contributions, the execution are all clearly described. There are a good set of experiments (even "too many metrics"), visualizations of the expected improvement in alignment of the visual and audio token distributions, and overall the application of MMD or OT learning improves the alignment as demonstrated by the experimental results over the dominant baseline approach of contrastive learning. Reviewers agree that the paper puts together existing components in a novel and interesting way and that the cross-modal learning problem is relevant and deserves more work, since there is a significant gap in investment between image and audio modalities. 3 reviewers recommend a weak accept of this paper after the rebuttal phase (which resolved some initial questions) and the paper was discussed thoroughly, which would allow authors to prepare an improved camera-ready version.

On the flip side, the work may not be very impactful, since:
- the target audience may be small: it's not clear if the approach could be applied to other domains) since the task definition is very narrow and leaves out comparisons with other recent and coming LLMs. though the task is practically relevant (as confirmed by one of the reviewers), given current trends, other LLMs may quickly raise the bar, reducing the potential impact of this paper in practice.
- the performance is good on the "in-domain" task, but drops significantly on out-of-domain tasks (other databases). while the authors' definition of "zero-shot" use is consistent with other previous work (and their evaluations are fair), readers would reasonably expect better performance on Clotho for a "zero-shot" approach than what the method presently delivers
- experimental results are somewhat mixed and require a lot of attention to detail in order to fully understand. authors were able to increase reviewers' appreciation for the work and its theoretical justification during rebuttal, but it seems that readers (even of an improved camera-ready version) may still have other questions which they would not be able to resolve without access to the authors explanations